# ATM: Adaptive Time Series Tokenization with Semantic Modeling

## Abstract

Recent advances in time series forecasting have achieved remarkable success, yet two critical challenges remain underexplored: the limitations of fixed-length patching strategies and the lack of semantic-level modeling. Fixed-length patches struggle to capture heterogeneous temporal patterns and often truncate temporal patterns, while existing methods largely rely on data-driven statistical patterns without semantic guidance. We propose Adaptive Time Series Tokenization with Semantic Modeling (ATM), a novel framework designed to address these issues. ATM introduces a Temporal Tokenization Module, which consists of two interrelated components: the Time Tokenizer, which adaptively partitions time series to preserve meaningful patterns (e.g., full cycles or peaks), and the Semantic Tokenization Regularization, designed to ensure semantically coherent temporal partitioning. In addition, ATM incorporates Semantic-Aware Modeling, where a Semantic Extractor enriches patches with latent semantic information and a Semantic Modeler captures hierarchical dependencies from local temporal patches to global sequence structures, enhanced by a Mixture-of-Experts module for diverse pattern modeling. Extensive experiments show that ATM consistently surpasses state-of-the-art methods in both long-term and short-term forecasting, demonstrating its effectiveness and strong generalization ability. Code is available at `https://anonymous.4open.science/r/ATM`.

## 1 Introduction

In recent years, with the rapid development of deep learning techniques, a wide range of time series forecasting models have been proposed (Kim et al., 2025; Cho & Lee, 2025; Huang et al., 2025; Liu et al., 2022). Among them, patch-based methods (Yang et al., 2023) and Transformer-based (Wang et al., 2024b) architectures have been widely applied. The former enhances the model's ability to capture local structural patterns in a high-dimensional representation space (Chen et al., 2024; Liu et al., 2025b). By dividing time series into multiple patches, the model can extract richer temporal dependencies than those captured at individual time steps (Nie et al., 2023). The latter (Vaswani et al., 2017), owing to its powerful semantic modeling capabilities, has achieved remarkable success in natural language processing (Brown et al., 2020) and computer vision (Dosovitskiy et al., 2021), and has been extensively adapted for time series modeling tasks (Wu et al., 2021).

Despite the outstanding performance of existing methods (donghao & wang xue, 2024; Kong et al., 2025), two critical issues remain underexplored: (1) the limitations of fixed-length patching strategies and (2) the lack of semantic modeling for time series data.

First, most existing patch-based methods adopt a *fixed-length* patching strategy (Nie et al., 2023), which exhibits significant limitations when applied to complex time series data. Time series often exhibit heterogeneous patterns across multiple scales, including smooth long-term trends and abrupt local spikes. As a result, fixed-length patches cannot capture this diversity effectively: short patches overlook long-term dependencies, whereas long patches fail to capture sudden local changes. Moreover, fixed-length patching often truncates temporal patterns—for instance, a peak may be split across adjacent patches, disrupting its structural continuity. These issues ultimately limit the model's ability to capture rich and diverse temporal dynamics.

Second, semantic information plays a key role in revealing diverse temporal patterns in time series. For example, although certain fluctuations and anomalies may appear alike within temporal patches,

their semantic differences can drive divergent forecasted behaviors. However, existing methods generally lack semantic-level guidance during representation learning, relying on data-driven statistical patterns. As a result, models lack the ability to distinguish time-series patterns that exhibit numerical resemblance but semantic divergence, which leads to conflating distinct dynamics and undermines the modeling of temporal diversity.

Motivated by these challenges, we propose a novel adaptive time series modeling framework ATM. ATM introduces a Time Tokenizer (TT) module that dynamically partitions time series, thereby preventing the loss of complete temporal pattern. Moreover, we innovatively design a Semantic Tokenization Regularization (STR) mechanism for guidance using semantic information to ensure semantically coherent patching. To model temporal semantics, ATM integrates Semantic-Aware Modeling, comprising a Semantic Extractor and a Semantic Modeler. The Semantic Extractor generates enriched semantic embeddings for each patch by capturing latent semantic structures beyond raw numerical values, thereby enabling the discrimination of numerically similar but semantically different temporal patterns. Building on these representations, the Semantic Modeler captures hierarchical temporal semantics, ranging from patch-level dependencies to sequence-level interactions, and incorporates a Mixture-of-Experts module to adaptively model heterogeneous temporal patch patterns. The main contributions of this paper are summarized as follows:

• We propose a Temporal Tokenization Module, that adaptively partitions time series into semantically coherent patches via a Time Tokenizer and a Semantic Tokenization Regularization, effectively mitigating the boundary issues and rigidity of fixed-length patching.

• We propose Semantic-Aware Modeling, a Semantic Extractor enriches patches with semantic information for disambiguating superficially similar patterns, and a Semantic Modeler captures hierarchical dependencies and diverse temporal dynamics.

• Extensive experiments on seven real-world datasets demonstrate that ATM consistently outperforms state-of-the-art methods on both long-term and short-term forecasting tasks, verifying its effectiveness and generalization capability.

## 2 RELATED WORK

### 2.1 TIME SERIES FORECASTING WITH PATCH

In recent years, patch-based methods (Nie et al., 2023; Luo & Wang, 2023) have received increasing attention in time series forecasting. Compared to traditional approaches (Zhang & Yan, 2023; Lin et al., 2024) that do not utilize patches, these models demonstrate significant performance improvements, highlighting the importance of enhancing local information through patching. For instance, PatchTST (Nie et al., 2023) divides each time series into multiple patches and employs a self-attention mechanism to model dependencies across the temporal dimension, achieving state-of-the-art predictive performance. Following the success of PatchTST, recent studies such as PatchMLP (Tang & Zhang, 2025) and AMD (Han et al., 2024) have also adopted patch-based strategies to enrich sequence representations and further improve downstream task performance.

However, all these methods overlook the challenges associated with fixed patch lengths, which can lead to difficulties in modeling heterogeneous temporal patterns and to the truncation of temporal structures. Although methods like HDMixer (Huang et al., 2024) use extendable patches, their fixed number of patches prevents full coverage of the time series, leaving some time steps unrepresented. In this paper, we introduce the Temporal Tokenization Module to overcome the limitations of fixed-length patching strategies. This module can adaptively adjust both the length and the number of patches, thereby enhancing the usability of local semantic information within patches.

### 2.2 SEMANTIC MODELING

Currently, semantic modeling in time series forecasting primarily appears in studies integrating large language models (LLMs) (Hu et al., 2025; Pan et al., 2024; Jia et al., 2024; Wang et al., 2025). For instance, Time-LLM (Jin et al., 2024) leverages a cross-attention mechanism to align time series data with natural language modalities (mapping continuous signals to discrete representations), and then utilizes LLMs for temporal prediction. TimeCMA (Liu et al., 2025a) introduces a cross-modal

alignment module that retrieves decoupled and robust time series embeddings from LLM-enhanced prompt embeddings by computing inter-channel similarities, thereby enhancing forecasting capability. STEM-LTS (Zhao et al., 2025) designs a semantic-temporal alignment mechanism that strengthens the understanding and predictive ability of LLMs for time series data, effectively bridging the gap between numerical patterns and semantic representations (Chang et al., 2023).

Nevertheless, while LLMs provide powerful semantic modeling capabilities, their high computational and resource costs remain a major concern (Mohammed & Kora, 2025; Moradi et al., 2025). In contrast, methods without LLM integration often lack semantic modeling mechanisms (Xie et al., 2025), restricting their ability to generalize in scenarios requiring semantic discrimination (Sun et al., 2024). Therefore, we propose Semantic-Aware Modeling, comprising a Semantic Extractor that enriches patches with semantic information for distinguishing superficially similar patterns, and a Semantic Modeler that captures both hierarchical dependencies and diverse temporal dynamics.

## 3 METHODOLOGY

In multivariate time series forecasting, given the historical observations $\mathbf{X} = \{\mathbf{x}_1, \mathbf{x}_2, \ldots, \mathbf{x}_L\} \in \mathbb{R}^{L \times C}$, where $L$ denotes the number of time steps and $C$ denotes the number of features (i.e., channels). The model aims to predict the values for the next $S$ time steps $\mathbf{Y} = \{\mathbf{x}_{L+1}, \mathbf{x}_{L+2}, \ldots, \mathbf{x}_{L+S}\} \in \mathbb{R}^{S \times C}$.

### 3.1 STRUCTURE OVERVIEW

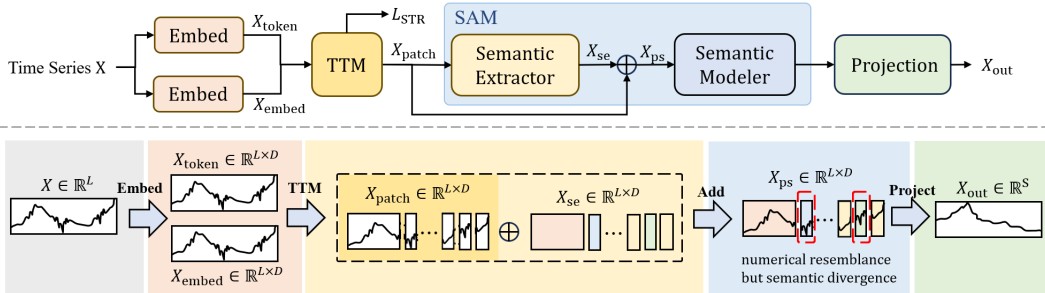

Figure 1: Overall structure and data flow of the proposed ATM.

The overall architecture of ATM is illustrated in Figure 1. The framework includes two main modules: the Temporal Tokenization Module (TTM) and the Semantic-Aware Modeling (SAM) Module. Based on our channel-independence strategy(Nie et al., 2023), we omit the channel dimension $C$ throughout the presentation for clarity.

The TTM is responsible for partitioning the input sequence into semantically coherent patches. It includes two components: the Time Tokenizer (TT) and the Semantic Tokenization Regularization (STR). First, the input time series $\mathbf{X}$ is linearly projected into two independent D-dimensional representations, denoted as $\mathbf{X}_{\text{token}} \in \mathbb{R}^{L \times D}$ and $\mathbf{X}_{\text{embed}} \in \mathbb{R}^{L \times D}$. These two representations are then fed into the TTM, which adaptively partitions the sequence into patches, formulated as:

$$\mathbf{X}_{\text{patch}} = \text{TTM}(\mathbf{X}_{\text{token}}, \mathbf{X}_{\text{embed}}). \tag{1}$$

Here, $\mathbf{X}_{\text{token}}$ is fed into the TT to capture the structural features of the input time series, and these features are then used to adaptively partition $\mathbf{X}_{\text{embed}}$ into patches of variable lengths and numbers.

However, because purely temporal partitioning often fails to capture semantic meaning, we introduce the Semantic Tokenization Regularization (STR) loss, denoted as $L_{\text{STR}}$. This loss leverages semantic cues $\mathbf{X}^{\text{sem}}$, provided by the Semantic Extractor during training, to enforce semantic coherence in the tokenization process. As shown in Figure 1, STR acts as an auxiliary constraint: it does not alter the tokenized outputs $\mathbf{X}_{\text{patch}}$, but ensures that the partitioning aligns with semantic structures.

The tokenized patches $\mathbf{X}_{\text{patch}}$ are then passed into the Semantic-Aware Modeling (SAM) Module, the second core component of ATM. The SAM module consists of a Semantic Extractor and a Semantic

Modeler. Within the Semantic Extractor, the semantic information of each patch is extracted and denoted as $\mathbf{X}_{\text{sem}} \in \mathbb{R}^{1 \times D}$. It is then linearly mapped to produce a semantic embedding $\mathbf{X}_{\text{se}} \in \mathbb{R}^{1 \times D}$, which is added to $\mathbf{X}_{\text{patch}}$ to enhance semantic information. This process can be formulated as:

$$\begin{aligned} \mathbf{X}_{\text{sem}} &= \text{SemanticExtractor}(\mathbf{X}_{\text{patch}}), \\ \mathbf{X}_{\text{se}} &= \text{Linear}(\mathbf{X}_{\text{sem}}), \\ \mathbf{X}_{\text{ps}} &= \mathbf{X}_{\text{patch}} + \mathbf{X}_{\text{se}}. \end{aligned} \quad (2)$$

Subsequently, $\mathbf{X}_{\text{ps}}$ is fed into Semantic Modeler. This module is composed of $N$ stacked blocks, where $N$ denotes the number of blocks. It is designed to model dependencies ranging from patch-level to sequence-level, as well as diverse patch patterns. Finally, the learned representations are projected through a prediction head to produce the output sequence $\mathbf{X}_{\text{out}} \in \mathbb{R}^S$.

## 3.2 TEMPORAL TOKENIZATION MODULE

The Temporal Tokenization Module (TTM) adaptively partitions time series into semantically coherent patches to preserve complete temporal patterns and better capture heterogeneous temporal dynamics.

### 3.2.1 TIME TOKENIZER

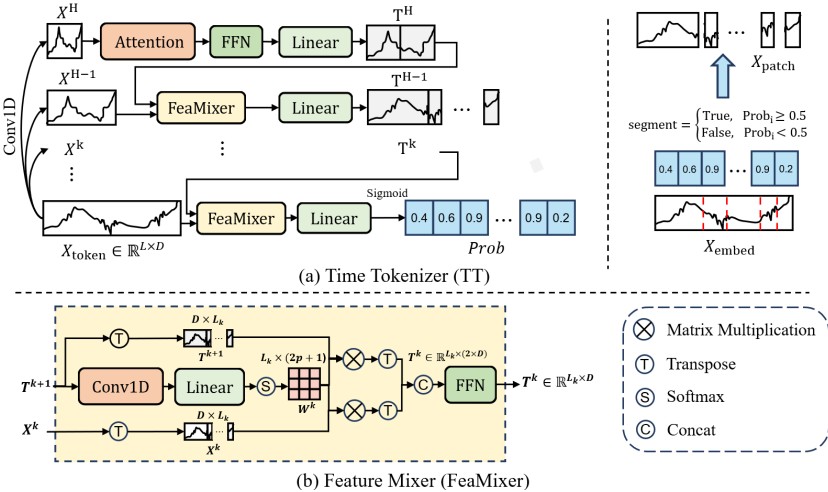

(a) Time Tokenizer (TT)

(b) Feature Mixer (FeaMixer)

Figure 2: Structure of Time Tokenizer (TT).

In time series modeling, most existing patching methods (e.g., the fixed-length strategy used in PatchTST) lack the flexibility to capture the heterogeneous temporal dynamics. A fixed-length segmentation strategy often truncates semantically coherent temporal patterns, since salient structural changes—such as peaks, abrupt fluctuations, or cycle boundaries—can occur at arbitrary positions and risk being split across patches. Such fragmentation undermines the continuity and integrity of temporal dependencies. This motivates a learnable adaptive partitioning mechanism, realized by the Time Tokenizer (TT), to automatically infer segment points according to temporal characteristics.

To mitigate the truncation of temporal segments, it adopts a hierarchical multi-granularity approach, first modeling coarse global structures and then refining boundaries at finer levels. Specifically, TT learns temporal features and estimates a boundary score for each time step, which is then used to identify semantically meaningful segment points, as illustrated in Figure 2.

The input $\mathbf{X}_{\text{token}}$ is first encoded by multi-scale 1D convolutions with exponentially increasing kernel sizes of $2^k (k = 1, 2, \ldots, H)$, producing representations at different granularities:

$$\mathbf{X}^k = \text{Conv1D}(\mathbf{X}_{\text{token}}). \quad (3)$$

At the coarsest granularity layer $H$, the input $\mathbf{X}^H$ is first processed by a multi-head self-attention and a feed-forward network (FFN), to obtain the initial patch divisions. The output is then projected

into the finer granularity space of layer $H-1$, yielding $\mathbf{T}^H$, which is then used by layer $H-1$ to further refine temporal boundaries at a finer scale.

At progressively finer levels $k$ ($k = 1, \ldots, H-1$), the $k$-th layer's input $\mathbf{X}^k$ and the $(k+1)$-th layer's output $\mathbf{T}^{k+1}$ are fed into the Feature Mixer module to perform cross-granularity feature fusion. Specifically, $\mathbf{T}^{k+1}$ is first passed through a 1D convolution with kernel size $2p+1$ and stride 1. For each time step $i$, this convolution extracts temporal features from its local neighborhood $[i - p, i + p]$, limited to the same patch from the previous $(k+1)$-th layer. Subsequently, the convolution output is projected and normalized with a Softmax layer to compute the importance weights of each neighboring time step relative to position $i$:

$$
\begin{aligned}
\mathbf{Z}^k &= \text{Conv1D}(\mathbf{T}^{k+1}), \\
\mathbf{W}_i^k &= \text{Softmax}(\text{Linear}(\mathbf{Z}_i^k)).
\end{aligned}
\tag{4}
$$

Based on the aforementioned importance weights, both coarse-grained and fine-grained information are selectively aggregated within the local neighborhood $[i - p, i + p]$:

$$
\begin{aligned}
\hat{\mathbf{T}}_i^{k+1} &= \mathbf{W}_i^k * \mathbf{T}_{i-p:i+p}^{k+1}, \\
\hat{\mathbf{X}}_i^k &= \mathbf{W}_i^k * \mathbf{X}_{i-p:i+p}^k.
\end{aligned}
\tag{5}
$$

The selected representations are then concatenated along the embedding dimension and fused through an FFN:

$$
\mathbf{T}^k = \text{FFN}(\text{Concat}(\hat{\mathbf{T}}^{k+1}, \hat{\mathbf{X}}^k)).
\tag{6}
$$

After recursive refinement, the final representation $\mathbf{T}^1$ produces boundary probabilities via a Sigmoid function, which determine the adaptive partitioning of $\mathbf{X}_{\text{embed}}$.

## 3.3 SEMANTIC TOKENIZATION REGULARIZATION

While the TT adaptively partitions the time series, it may still produce patches with inconsistent or noisy boundaries. To overcome this, we propose Semantic Tokenization Regularization (STR), which encourages the model to generate stable and representative temporal patterns. In this way, the resulting patches correspond to meaningful recurring behaviors rather than arbitrary fragments.

To encourage semantic coherence in partitioning, we introduce a clustering-based regularization on the semantic tokens $\mathbf{X}_{\text{sem}}$ extracted by the Semantic Extractor. Given a similarity threshold $r$, two semantic tokens $\mathbf{x}_i$ and $\mathbf{x}_j$ are considered similar if $d(\mathbf{x}_i, \mathbf{x}_j) \geq r$, where $d$ is a similarity function. For each unassigned token $\mathbf{x}_i \in \mathcal{U}$, we compute:

$$
n_i = |\{\mathbf{x}_j \in \mathcal{U} \mid d(\mathbf{x}_i, \mathbf{x}_j) \geq r\}|,
\tag{7}
$$

which counts the number of unassigned tokens similar to $\mathbf{x}_i$. Tokens with the largest $n_i$ are selected to form clusters with their similar neighbors, and the process is repeated until all tokens are assigned.

After clustering, Semantic Tokenization Regularization is defined as:

$$
\begin{aligned}
L_{\text{STR}} &= -\log(\bar{n}/N), \\
\bar{n} &= \frac{1}{c}\sum_{i=1}^{c} n_i,
\end{aligned}
\tag{8}
$$

where $c$ denotes the number of categories and $N$ denotes the number of tokens, respectively.

A key limitation of the above-mentioned STR loss is that it implicitly biases the model toward enlarging high-frequency categories, while neglecting low-frequency ones. This is problematic because low-frequency categories often correspond to rare but semantically critical patterns in time series (e.g., anomalies). To mitigate this issue, we refine the STR by applying it only to high-frequency categories, thereby preventing the loss from overwhelming rare but important categories. Concretely, we define a high-frequency threshold as:

$$
\begin{aligned}
r_h &= \bar{n} + k\sigma, \\
\sigma &= \sqrt{\frac{1}{c}\sum_{i=1}^{c}(n_i - \bar{n})^2},
\end{aligned}
\tag{9}
$$

where $\bar{n}$ and $\sigma$ denote the mean and standard deviation of token counts across all categories, and $k$ is a hyperparameter controlling the strictness of the threshold. Intuitively, only categories with counts significantly exceeding $r_h$ are considered "high frequency" and included in the STR computation. This adjustment allows STR to focus on frequent patterns without suppressing rare but critical ones.

## 4 SEMANTIC-AWARE MODELING

The framework leverages latent semantic information to ensure coherent temporal partitioning and to better capture diverse semantic structures.

### 4.1 SEMANTIC EXTRACTOR

The Semantic Extractor module extracts semantic tokens $\mathbf{X}_{\text{sem}}$ from $\mathbf{X}_{\text{patch}}$ and generates semantic embeddings $\mathbf{X}_{\text{se}}$, which are incorporated into the patch representation, as shown in Figure 3(a). The $\mathbf{X}_{\text{se}}$ enriches each patch with latent semantic information while simultaneously distinguishing the relative positions of different patches.

Specifically, for each patch, we prepend a learnable semantic query vector $\mathbf{q} \in \mathbb{R}^{1 \times D}$, which serves as its global semantic representation. By computing attention weights between this semantic query $\mathbf{q}$ and the patch representation $\mathbf{p}$, the semantic token of the patch $\mathbf{X}_{\text{sem}} \in \mathbb{R}^{1 \times D}$ is extracted.

However, a single semantic query lacks the capacity to capture the semantic diversity inherent in patch representations. To capture semantic diversity, we employ $Q$ independent query

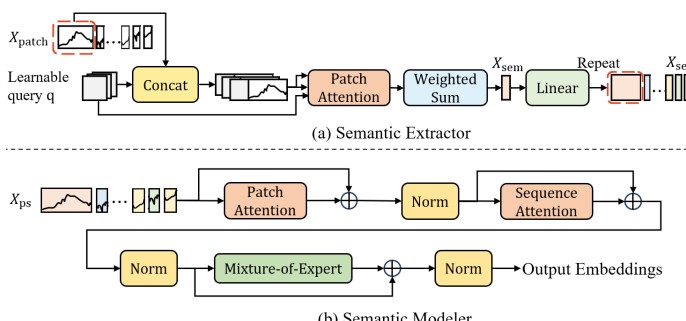

(a) Semantic Extractor

(b) Semantic Modeler

Figure 3: Structure of Semantic Extractor and Semantic Modeler

vectors $\{\mathbf{q}_1, \mathbf{q}_2, \ldots, \mathbf{q}_Q\}$, each attending to a different semantic aspect of the patch. The extracted tokens $\mathbf{X}_{\text{sem},k}(k = 1, \ldots, Q)$ are then fused via weighted aggregation into the final semantic representation:

$$r_k = \text{Linear}(\mathbf{X}_{\text{sem},k}),$$
$$\mathbf{X}_{\text{sem}} = \sum_{k=1}^{Q} r_k \, X_{\text{sem},k}, \tag{10}$$

where $r_k$ is the learnable weight for the $k$-th semantic token. The weighted sum is projected through a linear transformation to obtain the semantic embedding $\mathbf{X}_{\text{se}}$, which is added to the patch representation to jointly encode semantic and positional information.

### 4.2 SEMANTIC MODELER

As shown in Figure 3(b), the Semantic Modeler contains three layers designed to capture patch-level semantics, sequence-level semantics, and diverse patch patterns, respectively.

The first layer employs a patch-masked self-attention, constraining each time step to interact only with other time steps within the same patch. The self-attention score matrix is defined as:

$$\mathbf{A}_{ij} = \begin{cases} a_{ij}, & i, j \in P_i, \\ 0, & \text{otherwise.} \end{cases}, \tag{11}$$

where $a_{ij}$ denotes the attention score of the $j$-th time step on the $i$-th time step, and $P_i$ represents the patch to which the $i$-th time step belongs. The attention output is subsequently processed with residual connections and layer normalization to improve training stability and convergence speed.

The second layer utilizes standard multi-head self-attention to capture global semantic dependencies along the temporal dimension.

Considering that a single FFN has limited capacity to model diverse temporal patch patterns, we introduce the Mixture-of-Experts module (Shi et al., 2025) to enhance modeling capability. This module contains several expert networks, each like a standard FFN, specialized for different patch patterns. A routing strategy directs each time step to one or more experts.

## 5 OPTIMIZATION OBJECTIVES

The total loss during training is the weighted summation of the forecasting loss $L_{\text{fore}}$ and the Semantic Tokenization Regularizatio loss $L_{\text{STR}}$:

$$L_{\text{total}} = L_{\text{fore}} + \lambda L_{\text{STR}}, \tag{12}$$

where $\lambda$ denotes the weight of $L_{\text{STR}}$.

## 6 EXPERIMENT

To demonstrate the effectiveness of the proposed ATM, we conduct extensive experiments on long-term forecasting and short-term forecasting.

**Baselines.** We select representative baselines from recent advances in time series forecasting, covering the following categories: (1) Transformer-based models: Times2D (Nematirad et al., 2025), TQNet (Lin et al., 2025), iTransformer (Liu et al., 2024), PatchTST (Nie et al., 2023), ETSformer (Woo et al., 2022) and FEDformer (Zhou et al., 2022); (2) LLMs-based models: CALF (Liu et al., 2025c), TimeLLM (Jin et al., 2024), GPT4TS (Zhou et al., 2023); (3) MLP-based models: AMD (Han et al., 2024), HDMixer (Huang et al., 2024), DLinear (Zeng et al., 2023) and TiDE (Das et al., 2024); (4) CNN-based models: TimesNet (Wu et al., 2023), MICN (Wang et al., 2023).

**Implementation Details.** Our model is optimized with Adam optimizer (Kingma & Ba, 2017). For long-term forecasting tasks, the L1 loss is employed across all datasets. For short-term forecasting, the model is refined using the SMAPE loss. Additionally, we adopt a random seed of 2025 to ensure reproducibility. All our training processes are conducted on a single RTX 4090 GPU.

### 6.1 LONG-TERM FORECASTING

**Setups.** We conduct experiments on six widely-used real-world datasets, including four ETT datasets (ETTh1, ETTh2, ETTm1, ETTm2), Weather, and Electricity. The input time series length $L$ is fixed as 96 for a fair comparison, and we adopt four distinct prediction horizons $H \in \{96, 192, 336, 720\}$. Consistent with prior works (Wang et al., 2024a), the mean squared error (MSE) and mean absolute error (MAE) are employed as evaluation metrics.

**Results.** Comprehensive long-term forecasting results are presented in Table 1. ATM demonstrates state-of-the-art performance, achieving the best results in 48 evaluation tasks, whereas the closest competing baseline achieves the best results in only 3 tasks. Notably, ATM achieves an average reduction of 4.66% in MSE and 4.08% in MAE compared to the state-of-the-art patch-based model PatchTST on the ETTh1 dataset. Compared with Transformer-based baselines, ATM achieves a 3.99% reduction in MSE on the ETTh2 dataset relative to TQNet, and yields reductions of 5.51% in MSE and 5.58% in MAE on the ETTh1 dataset relative to iTransformer.

### 6.2 SHORT-TERM FORECASTING

**Setups.** We employ the M4 datasets (Makridakis et al., 2018), comprising $100,000$ univariate marketing data with six frequencies from hourly to yearly. In this setting, the prediction horizons are relatively short, ranging in $[6, 48]$. Accordingly, the input lengths are configured to be twice the size of prediction horizons. The evaluation metrics are symmetric mean absolute percentage error (SMAPE), mean absolute scaled error (MASE), and overall weighted average (OWA).

**Results.** As shown in Table 2, ATM demonstrates superior performance in short-term forecasting across multiple evaluation metrics. Notably, it achieves the best results in 12 out of 15 cate-

Table 1: Long-term forecasting results. The best performance is highlighted in **red**, and the second-best is underlined.

| Models | | ATM | | Times2D | | TQNet | | iTransformer | | PatchTST | | CALF | | TimeLLM | | AMD | | HDMixer | | TimesNet | | MICN | |
|---|---|---|---|---|---|---|---|---|---|---|---|---|---|---|---|---|---|---|---|---|---|---|---|
| Metric | | MSE | MAE | MSE | MAE | MSE | MAE | MSE | MAE | MSE | MAE | MSE | MAE | MSE | MAE | MSE | MAE | MSE | MAE | MSE | MAE | MSE | MAE |
| ETTh1 | 96 | **0.369** | **0.385** | 0.371 | 0.409 | 0.372 | 0.391 | 0.386 | 0.404 | 0.394 | 0.407 | 0.370 | 0.388 | 0.397 | 0.408 | 0.370 | 0.397 | 0.372 | 0.398 | 0.383 | 0.403 | 0.418 | 0.429 |
| | 192 | **0.427** | **0.417** | 0.430 | 0.427 | 0.430 | 0.421 | 0.441 | 0.436 | 0.446 | 0.434 | 0.427 | 0.421 | 0.451 | 0.440 | 0.431 | 0.418 | 0.433 | 0.422 | 0.438 | 0.430 | 0.470 | 0.485 |
| | 336 | 0.460 | **0.436** | 0.464 | 0.439 | 0.475 | 0.438 | 0.486 | 0.457 | 0.485 | 0.451 | 0.455 | 0.436 | 0.505 | 0.470 | 0.462 | 0.438 | 0.465 | 0.439 | 0.491 | 0.467 | 0.570 | 0.551 |
| | 720 | **0.464** | **0.455** | 0.469 | 0.460 | 0.481 | 0.469 | 0.508 | 0.493 | 0.479 | 0.471 | 0.475 | 0.467 | 0.481 | 0.472 | 0.464 | 0.463 | 0.468 | 0.460 | 0.520 | 0.498 | 0.769 | 0.671 |
| | Avg | **0.430** | **0.423** | 0.434 | 0.434 | 0.440 | 0.430 | 0.455 | 0.448 | 0.451 | 0.441 | 0.432 | 0.428 | 0.459 | 0.448 | 0.432 | 0.429 | 0.435 | 0.430 | 0.458 | 0.450 | 0.557 | 0.534 |
| ETTh2 | 96 | **0.278** | **0.327** | 0.284 | 0.334 | 0.294 | 0.341 | 0.300 | 0.347 | 0.294 | 0.342 | 0.279 | 0.330 | 0.294 | 0.346 | 0.279 | 0.331 | 0.280 | 0.331 | 0.341 | 0.374 | 0.298 | 0.364 |
| | 192 | **0.357** | **0.378** | 0.360 | 0.379 | 0.365 | 0.390 | 0.379 | 0.398 | 0.378 | 0.395 | 0.357 | 0.381 | 0.383 | 0.397 | 0.358 | 0.383 | 0.360 | 0.381 | 0.400 | 0.411 | 0.440 | 0.454 |
| | 336 | 0.408 | 0.405 | 0.387 | 0.419 | 0.415 | 0.418 | 0.418 | 0.429 | 0.381 | 0.409 | 0.362 | 0.399 | 0.449 | 0.444 | 0.407 | 0.411 | **0.360** | **0.395** | 0.450 | 0.450 | 0.655 | 0.565 |
| | 720 | **0.402** | **0.425** | 0.410 | 0.428 | 0.431 | 0.442 | 0.428 | 0.443 | 0.412 | 0.433 | 0.404 | 0.427 | 0.429 | 0.444 | 0.404 | 0.431 | 0.404 | 0.429 | 0.460 | 0.467 | 0.949 | 0.713 |
| | Avg | 0.361 | **0.384** | 0.360 | 0.390 | 0.376 | 0.400 | 0.381 | 0.404 | 0.366 | 0.395 | 0.351 | 0.384 | 0.389 | 0.408 | 0.362 | 0.389 | 0.351 | 0.384 | 0.414 | 0.426 | 0.586 | 0.524 |
| ETTm1 | 96 | **0.319** | **0.349** | 0.325 | 0.350 | 0.323 | 0.349 | 0.339 | 0.370 | 0.321 | 0.360 | 0.321 | 0.349 | 0.360 | 0.383 | 0.324 | 0.352 | 0.322 | 0.351 | 0.335 | 0.374 | 0.319 | 0.363 |
| | 192 | **0.367** | **0.374** | 0.371 | 0.376 | 0.371 | 0.374 | 0.382 | 0.395 | 0.368 | 0.384 | 0.373 | 0.375 | 0.380 | 0.391 | 0.371 | 0.376 | 0.368 | 0.379 | 0.374 | 0.387 | 0.370 | 0.390 |
| | 336 | **0.397** | **0.394** | 0.408 | 0.400 | 0.405 | 0.399 | 0.418 | 0.419 | 0.399 | 0.402 | 0.407 | 0.398 | 0.415 | 0.413 | 0.406 | 0.398 | 0.403 | 0.395 | 0.411 | 0.412 | 0.406 | 0.425 |
| | 720 | **0.452** | **0.428** | 0.470 | 0.432 | 0.460 | 0.432 | 0.487 | 0.456 | 0.453 | 0.435 | 0.476 | 0.437 | 0.483 | 0.449 | 0.462 | 0.431 | 0.461 | 0.430 | 0.478 | 0.448 | 0.480 | 0.477 |
| | Avg | **0.383** | **0.386** | 0.394 | 0.390 | 0.390 | 0.389 | 0.407 | 0.410 | 0.385 | 0.395 | 0.394 | 0.390 | 0.410 | 0.409 | 0.391 | 0.389 | 0.389 | 0.389 | 0.400 | 0.405 | 0.394 | 0.414 |
| ETTm2 | 96 | **0.175** | **0.252** | 0.181 | 0.257 | 0.179 | 0.255 | 0.185 | 0.272 | 0.179 | 0.260 | 0.179 | 0.256 | 0.194 | 0.281 | 0.178 | 0.256 | 0.177 | 0.255 | 0.186 | 0.265 | 0.180 | 0.275 |
| | 192 | **0.239** | **0.296** | 0.242 | 0.300 | 0.245 | 0.298 | 0.253 | 0.313 | 0.249 | 0.306 | 0.241 | 0.297 | 0.257 | 0.318 | 0.242 | 0.297 | 0.243 | 0.301 | 0.258 | 0.310 | 0.306 | 0.374 |
| | 336 | **0.302** | **0.337** | 0.308 | 0.341 | 0.306 | 0.339 | 0.315 | 0.350 | 0.312 | 0.346 | 0.305 | 0.339 | 0.316 | 0.350 | 0.307 | 0.338 | 0.303 | 0.340 | 0.318 | 0.351 | 0.324 | 0.388 |
| | 720 | **0.403** | **0.395** | 0.403 | 0.395 | 0.404 | 0.395 | 0.413 | 0.406 | 0.405 | 0.396 | 0.403 | 0.396 | 0.419 | 0.410 | 0.404 | 0.397 | 0.404 | 0.402 | 0.408 | 0.403 | 0.500 | 0.485 |
| | Avg | **0.279** | **0.320** | 0.284 | 0.323 | 0.284 | 0.322 | 0.292 | 0.335 | 0.286 | 0.327 | 0.282 | 0.322 | 0.297 | 0.340 | 0.283 | 0.322 | 0.282 | 0.325 | 0.293 | 0.332 | 0.327 | 0.381 |
| Weather | 96 | **0.165** | **0.202** | 0.168 | 0.205 | 0.165 | 0.204 | 0.173 | 0.214 | 0.176 | 0.218 | 0.165 | 0.204 | 0.195 | 0.231 | 0.167 | 0.203 | 0.167 | 0.203 | 0.171 | 0.220 | 0.166 | 0.229 |
| | 192 | **0.215** | **0.246** | 0.216 | 0.255 | 0.216 | 0.247 | 0.221 | 0.254 | 0.223 | 0.257 | 0.215 | 0.250 | 0.242 | 0.269 | 0.217 | 0.248 | 0.219 | 0.247 | 0.216 | 0.261 | 0.220 | 0.278 |
| | 336 | 0.270 | **0.287** | 0.271 | 0.292 | 0.272 | 0.289 | 0.278 | 0.296 | 0.278 | 0.297 | 0.270 | 0.290 | 0.295 | 0.304 | 0.265 | 0.293 | 0.273 | 0.289 | 0.281 | 0.305 | 0.275 | 0.331 |
| | 720 | 0.353 | 0.340 | 0.356 | 0.355 | 0.355 | 0.342 | 0.358 | 0.346 | 0.354 | 0.347 | 0.355 | 0.350 | 0.368 | 0.353 | 0.355 | 0.350 | 0.355 | 0.341 | 0.364 | 0.360 | **0.344** | 0.355 |
| | Avg | **0.251** | **0.269** | 0.253 | 0.277 | 0.252 | 0.271 | 0.258 | 0.278 | 0.258 | 0.280 | 0.252 | 0.274 | 0.275 | 0.289 | 0.251 | 0.274 | 0.254 | 0.270 | 0.258 | 0.287 | 0.251 | 0.298 |
| Electricity | 96 | 0.167 | 0.258 | 0.164 | 0.245 | **0.134** | 0.237 | 0.148 | 0.240 | 0.194 | 0.286 | 0.145 | 0.238 | 0.205 | 0.295 | 0.139 | 0.236 | 0.146 | **0.235** | 0.168 | 0.270 | 0.166 | 0.268 |
| | 192 | 0.174 | **0.260** | 0.171 | 0.267 | **0.154** | 0.261 | 0.162 | 0.267 | 0.199 | 0.291 | 0.161 | 0.261 | 0.206 | 0.292 | 0.158 | 0.270 | 0.164 | 0.273 | 0.184 | 0.290 | 0.174 | 0.287 |
| | 336 | 0.186 | **0.272** | 0.179 | 0.273 | 0.171 | 0.273 | 0.178 | 0.275 | 0.215 | 0.308 | 0.175 | 0.272 | 0.218 | 0.310 | **0.167** | 0.273 | 0.174 | 0.274 | 0.198 | 0.298 | 0.190 | 0.306 |
| | 720 | **0.220** | **0.304** | 0.227 | 0.315 | 0.223 | 0.305 | 0.225 | 0.317 | 0.256 | 0.332 | 0.222 | 0.304 | 0.263 | 0.340 | 0.224 | 0.305 | 0.223 | 0.305 | 0.221 | 0.319 | 0.223 | 0.321 |
| | Avg | 0.187 | 0.274 | 0.185 | 0.275 | **0.171** | 0.269 | 0.178 | 0.275 | 0.216 | 0.304 | 0.176 | **0.269** | 0.223 | 0.309 | 0.172 | 0.271 | 0.177 | 0.272 | 0.193 | 0.294 | 0.188 | 0.296 |
| 1st Count | | **48** | | 0 | | 3 | | 0 | | 0 | | 3 | | 0 | | 2 | | 3 | | 0 | | 1 | |

Table 2: Short-term forecasting results on M4 dataset. The input length and prediction length are set to [12, 96] and [6, 48], respectively.

| Models | | ATM | Times2D | PatchTST | ETSformer | FEDformer | GPT4TS | TimeLLM | DLinear | TiDE | TCN | MICN |
|---|---|---|---|---|---|---|---|---|---|---|---|---|
| Yearly | SMAPE | **13.331** | 13.416 | 13.476 | 18.007 | 13.727 | 13.530 | 13.418 | 16.966 | 15.318 | 14.918 | 25.020 |
| | MASE | **2.991** | 2.992 | 3.019 | 4.487 | 3.046 | 3.010 | 3.005 | 4.285 | 3.539 | 3.364 | 7.162 |
| | OWA | **0.784** | 0.785 | 0.791 | 1.115 | 0.803 | 0.790 | 0.788 | 1.058 | 0.910 | 0.879 | 1.668 |
| Quarterly | SMAPE | 10.267 | 10.364 | 10.382 | 13.376 | 10.793 | 10.178 | **10.118** | 12.144 | 11.831 | 11.120 | 15.217 |
| | MASE | 1.205 | 1.222 | 1.234 | 1.908 | 1.284 | 1.193 | **1.188** | 1.521 | 1.410 | 1.361 | 1.963 |
| | OWA | **0.904** | 0.915 | 0.923 | 1.305 | 0.958 | 0.906 | 0.905 | 1.107 | 1.052 | 1.001 | 1.407 |
| Monthly | SMAPE | **12.885** | 12.920 | 12.961 | 14.561 | 14.262 | 12.894 | 12.982 | 13.515 | 15.183 | 15.625 | 16.945 |
| | MASE | **0.975** | 0.979 | 0.975 | 1.370 | 1.105 | 0.976 | 0.978 | 1.037 | 1.191 | 1.274 | 1.443 |
| | OWA | **0.905** | 0.906 | 0.906 | 1.152 | 1.014 | 0.906 | 0.905 | 0.956 | 1.090 | 1.141 | 1.265 |
| Others | SMAPE | **4.938** | 4.950 | 4.955 | 7.269 | 4.957 | 4.942 | 4.939 | 6.713 | 6.120 | 7.181 | 41.985 |
| | MASE | **3.320** | 3.342 | 3.349 | 5.244 | 3.367 | 3.322 | 3.321 | 4.955 | 4.330 | 4.679 | 62.734 |
| | OWA | 1.040 | 1.047 | 1.048 | 1.593 | 1.039 | 1.025 | **1.005** | 1.488 | 1.330 | 1.494 | 14.313 |
| Average | SMAPE | **11.962** | 12.022 | 12.060 | 14.705 | 12.841 | 11.993 | 11.993 | 13.640 | 13.956 | 13.959 | 19.640 |
| | MASE | **1.611** | 1.618 | 1.626 | 2.410 | 1.707 | 1.613 | 1.612 | 2.096 | 1.941 | 1.946 | 5.948 |
| | OWA | **0.860** | 0.865 | 0.870 | 1.172 | 0.919 | 0.863 | 0.860 | 1.052 | 1.021 | 1.023 | 2.279 |
| 1st Count | | **12** | 0 | 0 | 0 | 0 | 0 | 3 | 0 | 0 | 0 | 0 |

gories, substantially outperforming all baselines. In comparison with PatchTST, the state-of-the-art patch-based method, ATM achieves an overall improvement of 1% in SMAPE. Compared with Transformer-based baselines, ATM achieves a 0.9% reduction in SMAPE relative to Times2D, and yields a reduction of 18.68% in SMAPE relative to ETSformer.

## 6.3 ABLATION STUDY

We conduct ablation studies on three datasets by removing (w/o) specific modules from ATM.

The results reported in Table 4 demonstrate that removing the Time Tokenizer leads to a substantial performance reduction, with average increases of 2.32%/2.15% in MSE/MAE on ETTh1. Incorporating STR yields a 3.34%/3.64% improvement in MSE/MAE on ETTm2. Similarly, the addition of Semantic Extractor yields a 4.07%/4.76% improvement in MSE/MAE on Weather. The best overall performance is achieved when all three modules are jointly employed, achieving the lowest MSE and MAE across all datasets.

Table 3: Ablation on different modules on ETTh1, ETTm2 and Weather datasets.

| Design | Time Tokenizer | STR | Semantic Extractor | ETTh1 | | ETTm2 | | Weather | |
|---|---|---|---|---|---|---|---|---|---|
| | | | | MSE | MAE | MSE | MAE | MSE | MAE |
| **ATM** | ✓ | ✓ | ✓ | **0.464** | **0.455** | **0.403** | **0.395** | **0.353** | **0.340** |
| | w/o | ✓ | ✓ | 0.475 | 0.465 | 0.411 | 0.403 | 0.361 | 0.350 |
| w/o | ✓ | w/o | ✓ | 0.473 | 0.462 | 0.417 | 0.410 | 0.360 | 0.348 |
| | ✓ | ✓ | w/o | 0.479 | 0.469 | 0.423 | 0.421 | 0.368 | 0.357 |

## 6.4 HYPERPARAMETER SENSITIVITY

In this section, we investigate the key hyperparameters of ATM, including the number of convolutional layers $H$ in the Time Tokenizer, the cosine similarity threshold $r$ and the $L_{STR}$ weight $\lambda$. The analysis is conducted on ETTh1 dataset with a fixed forecasting horizon of 192, where only one hyperparameter is varied while the others remain fixed. As shown in Figure 4, ATM performs better with the number of convolutional layers up to 3. When fewer layers are employed, the MAE increases, indicating that an insufficient number of convolutional layers is ineffective. In addition, intermediate values of both the cosine similarity threshold and STR weight provide better performance, whereas excessively large or small values lead to a gradual increase in loss.

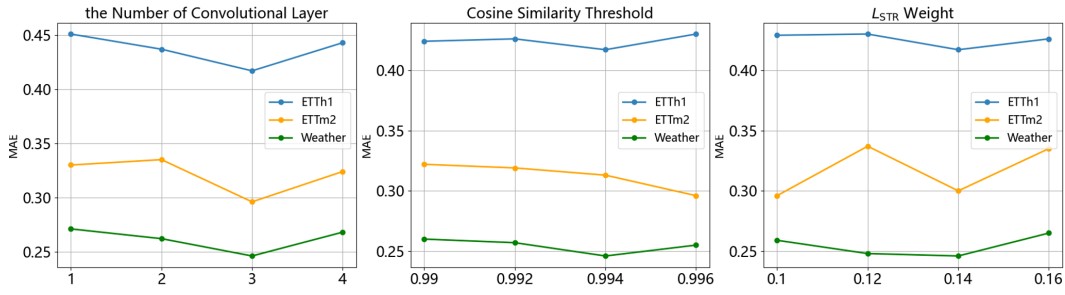

Figure 4: Sensitivity analysis of key hyperparameters for ATM with a prediction horizon of 192.

## 7 CONCLUSION

In this work, we address two key limitations in time series forecasting—the unreasonable fixed-length patching and the lack of semantic guidance—by proposing ATM, which leverages a Time Tokenizer for adaptive partitioning and a Semantic Tokenization Regularization to ensure semantically coherent patching. Furthermore, ATM incorporates Semantic-Aware Modeling with hierarchical dependency learning. Experiments show that ATM achieves state-of-the-art performance, highlighting its effectiveness and strong generalization.

## ETHICS STATEMENT

Our work exclusively uses publicly available benchmark datasets that contain no personally identifiable information. No human subjects are involved in this research.

## THE USE OF LARGE LANGUAGE MODELS

We promise not to use Large Language Models in writing.

## REPRODUCIBILITY STATEMENT

The promise that all experimental results can be reproduced. We have released our model code in an anonymous repository: `https://anonymous.4open.science/r/ATM`.

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
