# OpenReview forum: "ATM: Adaptive Time Series Tokenization with Semantic Modeling"
_ICLR.cc/2026/Conference — ICLR 2026 Conference Withdrawn Submission_

### Official Review · Reviewer_Q5Hm · 2025-10-19

**Soundness:** 2
**Presentation:** 2
**Contribution:** 2
**Rating:** 4
**Confidence:** 4

**Summary:**

This paper focuses on designing a new adaptive patch method for time series forecasting, and proposes a Adaptive Time Series Tokenization with Semantic Modeling (ATM) model. Experiments are carried out based on seven real-world datasets.

**Strengths:**

1. Tokenization is a critical step in deep learning models for time series modeling. Thereby, the investigated problem is good and worth exploring.

2. Source codes are provided for reproducibility.

**Weaknesses:**

1. Some relevant works about adaptive patching are not discussed, e.g., [1]

[1]. DeformableTST: Transformer for time series forecasting without over-reliance on patching. NIPS 2024.

2. Some recent baselines are missed in experiments, e.g., Timemixer and Timemixer++.

3. The datasets are limited, maybe adding some traffic datasets, e.g., PeMS could make the experimental parts more solid.

**Questions:**

1. The author spent a considerable amount of space discussing the drawbacks of the fixed patch, but what are its advantages? Can it, to some extent, ensure that the information strength between tokens is consistent? This way, there won't be any deviation or collapse when calculating attention. If the varying-length patch method is used, then the amount of information carried by each token will be different. Will this lead to an information imbalance during the attention process?

2. In the field of vision, there are some methods involving deformable convolutions, which might be related to adaptive patches. Could the authors elaborate on further discussions and comparisons?

---

### Official Review · Reviewer_J1di · 2025-10-27

**Soundness:** 2
**Presentation:** 1
**Contribution:** 2
**Rating:** 2
**Confidence:** 4

**Summary:**

In this study, the authors investigate the tokenisation of time series in the context of forecasting. In particular, they propose to divide time series into (1) patches of varying length and (2) patches that are semantically meaningful. Experiments are conducted across 10 tasks, involving short-term and long-term forecasting. An ablation study is conducted to investigate the importance of the proposed components.

**Strengths:**

1) The paper is well structured.
2) The authors conduct forecasting experiments on established datasets, including 4 short-term and 6 long-term forecasting tasks, and an ablation study to evaluate their method.

**Weaknesses:**

Related Works:

1) The authors do not discuss the most recent works in the field of time series tokenisation. For instance, works on domain-specific tokenisation [1] and wavelet-based tokenisation [2] should be included to provide a representative overview of the current literature.

Presentation:

2) The authors propose ATM with its multiple components, however, the presentation lacks clarity. For instance, Figures 1 (overall architecture), 2 (Time Tokeniser), and 3 (Semantic Extractor and Semantic Modeler) are difficult to follow.

Experiments:

3) The authors compare the proposed tokeniser against several baselines, however, they do not include foundational models. Particularly, general-purpose models (e.g. OTIS [1] and MOMENT [3]) and foundational forecasting models (e.g. MOIRAI [4] and Chronos [5]) should be included as baselines to ensure a fair comparison of the proposed method.

4) As the authors only report scores for a single seed (2025), the study does not guarantee robustness of the results. Mean and standard deviation across multiple seeds should be reported in Tables 1, 2, and 3.

Reproducibility:

5) The authors do not support reproducibility by making their code publicly available for evaluation.

Discussion:

6) The discussion of the experiments is too limited. The experiments section only provides numbers without their interpretation, while the conclusion section only provides a brief summary of the abstract.
7) The authors do not discuss the limitations of their work.

___
[1] Turgut, Özgün, et al. "Towards generalisable time series understanding across domains." arXiv preprint arXiv:2410.07299 (2024).

[2] Masserano, Luca, et al. "Enhancing Foundation Models for Time Series Forecasting via Wavelet-based Tokenization." ICML (2025).

[3] Goswami, Mononito, et al. "MOMENT: A Family of Open Time-series Foundation Models." ICML (2024).

[4] Woo, Gerald, et al. "Unified training of universal time series forecasting transformers." ICML (2024).

[5] Ansari, Abdul Fatir, et al. "Chronos: Learning the Language of Time Series." TMLR (2024).

**Questions:**

1) What is the motivation behind using patches of varying length?
2) How do the authors (a) define and (b) quantify semantic information within a patch? Does semantic information vary with respect to the time series domain? If this is the case, how does it affect the tokenisation across domains, e.g. for medical recordings and weather recordings?
3) What is the motivation behind using a SMAPE loss and L1 loss in short-term and long-term forecasting, respectively?
4) How do the authors determine $\lambda$ in Equation (12), which balances the $L_\text{fore}$ and $L_\text{STR}$ loss terms?
5) Have the authors tried using their approach to different types of tasks, such as classification or regression? For instance, domain-specific tokenisation proposed in [1] proves beneficial across classification, regression, and forecasing tasks, further advancing the field of time series analysis.

___
[1] Turgut, Özgün, et al. "Towards generalisable time series understanding across domains." arXiv preprint arXiv:2410.07299 (2024).

---

### Official Review · Reviewer_qdb5 · 2025-11-01

**Soundness:** 2
**Presentation:** 3
**Contribution:** 2
**Rating:** 4
**Confidence:** 5

**Summary:**

The paper proposes a framework that combines adaptive patching and semantic modeling to improve time series forecasting. It features a learnable patching module (TTM) that generates variable-length patches and a Semantic Tokenization Regularization (STR) mechanism to encourage semantic coherence. The model show strong empirical results across multiple benchmarks.

**Strengths:**

* The proposed approach is well-motivated. The goal of avoiding temporal truncation in fixed patching is valid and important.
* ATM achieves competitive performance across several datasets, showing clear gains over strong baselines like PatchTST.

**Weaknesses:**

* The paper overlooks highly relevant concurrent research such as TimeCAT (dynamic grouping), which target the same limitation of fixed patching, so the claim that this area is “underexplored” is misleading.
* The STR uses cosine similarity, a weak statistical proxy, and the paper fails to convincingly justify or validate that it captures genuine temporal **semantics**.

**Questions:**

* Why is TimeCAT (ICLR2025 submission) not cited or compared? Both address adaptive segmentation, so a direct comparison seems essential.
* How much of the performance gain comes from the SAM module (especially its MoE) versus the Time Tokenizer itself?
* Is there any experiment showing that STR captures real-world semantic differences rather than statistical similarity?

---

### Note · Authors · 2026-01-07

I have read and agree with the venue's withdrawal policy on behalf of myself and my co-authors.